# Exploration into Galectin-3 Driven Endocytosis and Lattices

**DOI:** 10.3390/biom14091169

**Published:** 2024-09-18

**Authors:** Massiullah Shafaq-Zadah, Estelle Dransart, Satish Kailasam Mani, Julio Lopes Sampaio, Lydia Bouidghaghen, Ulf J. Nilsson, Hakon Leffler, Ludger Johannes

**Affiliations:** 1Cellular and Chemical Biology Unit, Institut Curie, Paris Sciences & Lettres Research University, U1143 INSERM, UMR3666 CNRS, 75248 Paris, France; estelle.dransart@curie.fr (E.D.); satish.kailasam@ijm.fr (S.K.M.); 2CurieCoreTech–Metabolomics and Lipidomics Platform, Institute Curie, 75248 Paris, France; julio.lopes-sampaio@curie.fr (J.L.S.); lydia.bouidghaghen@curie.fr (L.B.); 3Department of Chemistry, Lund University, 221 00 Lund, Sweden; ulf.nilsson@chem.lu.se; 4Section MIG (Microbiology, Immunology, Glycobiology), Department of Laboratory Medicine, Lund University, 221 00 Lund, Sweden; hakon.leffler@med.lu.se

**Keywords:** endocytosis, galectin-3, glycosphingolipids, α_5_β_1_ integrin, gl-lect hypothesis, lattices, clathrin

## Abstract

Essentially all plasma membrane proteins are glycosylated, and their activity is regulated by tuning their cell surface dynamics. This is achieved by glycan-binding proteins of the galectin family that either retain glycoproteins within lattices or drive their endocytic uptake via the clathrin-independent glycolipid-lectin (GL-Lect) mechanism. Here, we have used immunofluorescence-based assays to analyze how lattice and GL-Lect mechanisms affect the internalization of the cell adhesion and migration glycoprotein α_5_β_1_ integrin. In retinal pigment epithelial (RPE-1) cells, internalized α_5_β_1_ integrin is found in small peripheral endosomes under unperturbed conditions. Pharmacological compounds were used to competitively inhibit one of the galectin family members, galectin-3 (Gal3), or to inhibit the expression of glycosphingolipids, both of which are the fabric of the GL-Lect mechanism. We found that under acute inhibition conditions, endocytic uptake of α_5_β_1_ integrin was strongly reduced, in agreement with previous studies on the GL-Lect driven internalization of the protein. In contrast, upon prolonged inhibitor treatment, the uptake of α_5_β_1_ integrin was increased, and the protein was now internalized by alternative pathways into large perinuclear endosomes. Our findings suggest that under these prolonged inhibitor treatment conditions, α_5_β_1_ integrin containing galectin lattices are dissociated, leading to an altered endocytic compartmentalization.

## 1. Introduction

Galectins are a family of 15 members of β-galactoside binding proteins [1,2]. They are synthetized in the cytosol and translocated to the extracellular space by unconventional secretion [3,4] to interact with glycans on proteins and lipids. In addition, galectins also interact intracellularly with many cytosolic and nuclear proteins in a glycan-independent manner [5]. Striking examples for cytosolic implications include galectin-3 (Gal3) association with endosomal sorting complex [6], β-catenin and Wnt signaling pathways [7], and the regulation of apoptosis [8]. Gal3 has also been reported to shuttle from the cytoplasm to the nucleus [9,10], where it has been linked with mRNA splicing and gene expression [11,12] and antiapoptotic activity [5]. More globally, many pathophysiological functions have been reported for galectins in the nucleus, the cytosol, and the extracellular space [13].

All galectins share a C-terminally located carbohydrate recognition domain (CRD) as a common feature. Gal3 stands out among the galectin family members by the presence of an additional unstructured N-terminal domain (Figure 1A, top) that is strongly involved in the capacity of the protein to form oligomers [14,15,16] (Figure 1A, bottom). This effect is possibly based on the N-terminal’s propensity to form biomolecular condensates [17,18,19]. Structural models of Gal3 oligomers have been proposed, including ill-defined pentamers [20], tetramers with inconsistent features made from N-terminally truncated Gal3 [21], and higher order assemblies [22].

Oligomerization-competent extracellular galectins have been implicated in the formation of extended lattices with glycoproteins whose lateral motility, membrane residency [23,24], and biological activities are thereby altered [20,25,26,27]. These include the EGF and TGF-β growth factor receptors [28,29], the T cell receptor [25,30], the metabolic glucose transporter Glut2, and the glucagon receptor [31,32].

Oligomerization-competent extracellular galectins have also been identified as drivers of clathrin-independent endocytosis, which has been termed the GlycoLipid-Lectin (GL-Lect) hypothesis [33]. Research leading up to the formulation of the GL-Lect hypothesis was based on pentameric pathogenic lectins from the bacterial Shiga and cholera toxins and from the polyoma virus SV40, which all hijack glycosphingolipids (GSLs) at the cell surface not only as cellular receptors but also to induce narrow plasma membrane bending resulting in the formation of tubular endocytic pits [34,35,36,37,38]. Curvature is induced by a specific arrangement of glycan binding sites on the oligomeric lectins [33] and by the fact that these oligomeric lectins develop the capacity to compress lipids in the exoplasmic leaflet [39]. These tubular endocytic pits are then further processed by cytosolic machinery such as actin [38], scaffolding BAR-domain proteins, and molecular motors to initiate friction-driven scission [36,40,41]. Tubular and crescent-shaped clathrin-independent endocytic carriers (CLICs) are thereby generated [42,43].

Possibly the most important functional hallmark of the above-mentioned oligomeric pathogenic lectins is their capacity to transform flat membranes into tubular membrane invaginations in interaction with GSLs [34,37,44]. It was therefore of high interest when it was found that Gal3 has the same GSL and oligomerization-dependent membrane bending activity [45]. Importantly, it then turned out that the endocytic uptake of cellular proteins such as α_5_β_1_ integrin and CD44 is driven by Gal3 in a GSL-dependent manner via CLICs [42,45,46], thereby identifying these as endogenous GL-Lect cargoes. GL-Lect driven endocytosis appears to operate more generally since Gal4 is also found within CLIC structures [45] and Gal8 drives the clathrin-independent endocytosis of the cell adhesion protein CD166 [47].

GSLs are the most abundant glycosylated lipids found in vertebrates [48]. GSLs belong to a class of heterogenous amphipathic lipids featuring complex glycan structures that face the extracellular milieu and that are linked via the β-glycosidic bond to a ceramide backbone embedded in the lipid bilayer [49]. Of importance, the functions of GSLs as pathogen receptors and in the internalization of plasma membrane proteins have been reported [33]. Striking examples include Shiga and cholera toxins as well as simian virus, all of which depend on GSLs to achieve efficient binding and subsequent internalization to induce their pathogenicity [49,50,51,52] (see above). Genetic depletion of GSL expression in murine intestine compromises nutrient resorption by yet unidentified receptors and leads to lethality [53,54]. Recently, the cellular uptake of key cell adhesion molecules such as CD44, the receptor of the glycosaminoglycan hyaluronic acid, and α_5_β_1_ integrin have been reported to depend on GSL expression [45,55].

α_5_β_1_ integrin is a highly glycosylated heterodimeric protein that, in interaction with the extracellular matrix component fibronectin, is essential for cell adhesion and migration [56,57,58]. α_5_β_1_ integrin is also a well-established glycan-dependent interacting partner of Gal3 [45,46,59,60]. Of note, glycosylation plays a central role in the biological activities of integrins [59,61,62,63,64,65,66,67,68,69,70]. For its endocytic uptake into cells, α_5_β_1_ integrin not only uses the aforementioned clathrin-independent endocytic mechanisms, but also the well characterized clathrin-dependent pathway [45,46,71,72,73,74,75,76,77,78,79].

Here, we have used α_5_β_1_ integrin as a model to address the apparent contradiction between galectin lattices as a means to retain cargo proteins at the cell surface, and GL-Lect driven endocytosis as a mechanism to internalize these cargo proteins into cells. Our findings indicate that Gal3 and α_5_β_1_ integrin colocalize at the cell surface in clusters of variable size and shape. The acute incubation of cells with small molecule inhibitors of Gal3 or of GSL expression results in a significant reduction of α_5_β_1_ integrin uptake, which agrees with previous findings [45,46]. In contrast, prolonged incubation with these inhibitors leads to an increased cargo uptake rate. Based on the analysis of the intracellular distribution of internalized α_5_β_1_ integrin and on functional experiments, we suggest that prolonged inhibitor treatment liberates α_5_β_1_ integrin from galectin lattices. With GL-Lect driven endocytosis being inhibited, uptake now occurs massively via the clathrin and macropinocytosis pathways into endosomes that are different from the ones that are targeted under unperturbed conditions. We discuss how lattice and GL-Lect mechanisms may thereby operate as a continuum at the cell surface.

## 2. Materials and Methods

***Cells:*** Human retinal pigment epithelial RPE-1 cells from ATCC (hTERT RPE-1, Ref. CRL-4000).

***Reagents:*** Anti-β_1_ mAb13 antibody (BD Bioscience, Le Pont de Claix, France; Ref. 552828), anti-Gal3 antibody (Fu-Tong Liu, UC Davis, CA, USA), Cy3-labeled anti-β_1_ integrin mAb13 antibody, anti-EEA1 antibody (Biorbyt, Durham, US; Ref. 11606), Cy3-labeled donkey anti-rat antibody (Beckman Coulter, Roissy CDG, France; Ref. 712-166-153), Cy3-labeled anti-goat antibody, fluorescein-isothiocyanate dextran 70 kDa (Sigma Merck, Saint Quentin Fallavier, France; Ref. 46945), Gal3 inhibitor compound I3 (1,1′-sulfanediyl-bis-{3-deoxy-3-[4-(butylaminocarbonyl)-1H-1,2,3-triazol-1-yl]-β-D galactopyranoside} [80,81], Genz-123346 (Sigma Merck, Saint Quentin Fallavier, France; Ref. 5382850001), β-D-lactose (Sigma Merck, Saint Quentin Fallavier, France; Ref. L3750), NHS-Cy3 (GE, Buc, France; PA23001), NHS-CF680 (Sigma Merck, Saint Quentin Fallavier, France; SCJ4600055), transferrin-Alexa546 (Tf-A546) (Invitrogen, Saint Aubin, France; Ref. T23364), AlexaFluor488-labeled recombinant purified Gal3, Cy3-labeled recombinant purified Gal3, AlexaFluor647-labeled recombinant purified Gal3.

***Media and buffers:*** DMEM-F12 Gibco (Thermofisher, Villebon sur Yvette, France; Ref. 11320033) supplemented or not with 10% FBS, PBS (137 mM NaCl, 2.7 mM KCl, 8 mM Na_2_HPO_4_, and 2 mM KH_2_PO_4_) (Thermofisher, Villebon sur Yvette, France; ref. 14190144), PBS^++^ (PBS supplemented with 1 mM MgCl_2_ and 0.5 mM CaCl_2_, pH 7.4), ***β***-D-lactose wash solution (150 mM, iso-osmolarized in DMEM-F12) ( Sigma Merck, Saint Quentin Fallavier, France; Ref. L3750), I3 solution (10 mM, in DMSO), PFA 4% solution (Electron Microscopy Sciences, Hatfield, US; Ref. 1570), BSA buffer (0.2% BSA in PBS) (Euromedex, Souffelweyersheim, France), BSA-saponin buffer (0.5% saponin, 0.2% BSA, in PBS), acid wash buffer (glycine 0.5 M, pH 2.2).

***Equipment:*** Confocal microscope A1RHD25 (Nikon, Amstelveen, The Netherlands; Nikon Imaging Center, Institut Curie), Abbelight SAFe 360 microscope (Leica Microsystems, Nanterre, France)

***I3 treatment:*** For acute treatment, cells were pre-treated for 5 min at 37 °C with 10 μM I3 diluted in DMEM F12 (without FBS). I3 was washed out three times with the same medium and cargoes uptake was performed according to experimental conditions (see antibody, transferrin and Gal3 uptake sections). For prolonged treatment, the 5 min I3 pre-treatment (10 μM at 37 °C) was followed by 10 min incubation at 37 °C with the different aforementioned cargoes in DMEM F12 medium and in the continued presence of I3, resulting in a total incubation time of 15 min with I3.

***Genz treatment:*** RPE-1 cells were continuously incubated for three or five days with 5 μM Genz-123346 in DMEM-F12 medium containing 5% FBS for acute or prolonged treatments, respectively, prior to performing uptake assays as described below.

***siCHC transfection:*** Using HiPerFect reagent, RPE1 cells were transfected with 8 nM of siCHC (or of siCtrl) to inhibit clathrin heavy chain expression. Uptake was performed 72 h after transfection. Inhibition of CHC expression was assessed by immunoblotting. 

***Antibody uptake:*** 5 μg/mL of anti-β_1_ integrin (mAb13) antibody diluted in DMEM-F12 supplemented (Genz treatment) or not (I3 treatment) with 10% FBS were continuously incubated for 10 min at 37 °C with RPE-1 cells according to experimental conditions. Cells were shifted to 4 °C and the excess of antibodies removed by washing with ice-cold PBS^++^. Cell surface exposed antibodies were removed with three acid washes of 45 s each. Acidic pH was then neutralized with three-times ice-cold PBS^++^ washes, and cells were fixed in 4% PFA followed by immunofluorescence labeling as described in the immunofluorescence section.

***Antibody binding assay:*** RPE-1 cells were chilled on ice for 10 min. Then, 10 μg/mL of anti-β_1_ integrin antibody (mAb13) or 5 μg/mL of anti-Gal3 antibody were incubated for 30 min at 4 °C with cells. After three PBS^++^ washes, cells were fixed with 4% PFA followed by immunofluorescence labeling (see immunofluorescence section).

***Effect of I3 treatments on exogenously pre-bound Gal3:*** 1 μg/mL of Cy3-labeled Gal3 was incubated with pre-cooled RPE-1 cells (4 °C) for 30 min, followed by acute (5 min) or prolonged (15 min) incubations with I3 (10 μM), as described in the “I3 treatment” section. Cells were then fixed with 4% PFA. Coverslips were mounted on slides with Mowiol supplemented with DAPI. 

***anti-β_1_ integrin antibody labeling with NHS-CF680:*** mAb13 antibody was labeled for 2 h at 21 °C with a 10-molar excess of NHS-CF680 in PBS buffer. Unreacted fluorophore was quenched for 20 min at 21 °C with 20 mM Tris and then removed using 10 kDa cutoff desalting columns.

***Gal3 binding or Gal3 and anti-β_1_ integrin antibody co-binding for dSTORM analysis:*** RPE-1 cells grown on high resolution #1.5 glass coverslips (THOR labs) were chilled on ice for 10 min. Then, 200 nM of AlexaFluor647-labeled Gal3 were incubated for 30 min at 4 °C with these cells. After three PBS^++^ washes, cells were either directly fixed with 4% PFA (Gal3 binding only) or further incubated with 5 μg/mL of CF680-labeled anti-β_1_ integrin (mAb13) antibody for 30 min at 4 °C before 4% PFA fixation.

***Transferrin and Gal3 uptake:*** 5 μg/mL of Tf-A546 or 200 nM of Alexa488-Gal3 or Cy3-Gal3 diluted in DMEM F12 (without FBS) were continuously incubated for 10 min at 37 °C with RPE-1 cells, according to experimental conditions. For Gal3 uptake, cell surface bound Gal3 was removed by three lactose washes (150 mM).

***Dextran uptake:*** 10 mg/mL of FITC-dextran were continuously incubated for 10 min at 37 °C with RPE-1 cells, according to experimental conditions.

***Immunofluorescence:*** Cells were fixed for 5 min at 4 °C with 4% PFA and for an additional 5 min at room temperature. Excess of PFA was then quenched with 50 mM NH4Cl, followed by incubation for 30 min at room temperature with BSA-saponin saturation/permeabilization solution (intracellular immunostaining), or only for saturation with PBS, 0.2% BSA (plasma membrane labeling). Cells were incubated for 30 min at room temperature either with primary and secondary antibodies for the labeling of cellular antigens (EEA1), or only with secondary antibody to detect primary antibodies that had been put in contact with living cells (for antibody binding and uptake experiments). Of note, 1 h incubation was used with the primary anti-Gal3 antibody. Coverslips were mounted on slides with Mowiol supplemented with DAPI. All immunofluorescence images on fixed cells were acquired with a Nikon A1RHD25 confocal microscope. Signals were quantified using the ImageJ program and displayed as mean intensity per cell. For co-localization analysis, the JACoP plugin (version 2.1.1) in ImageJ (version 2.0.0-rc-69/1.52p) was used to measure Manders’ coefficient signal co-occurrences. This quantification method calculates the percentage of signal from one channel that overlaps with signal from the other channel. To do so, it is of importance to image samples with high signal to noise ratios, and to select pixels from both channels that are relevant from the biological perspective [82].

***Kinetics of Genz treatment and cell harvesting for lipidomics:*** RPE-1 cells were incubated either with 5 μM Genz-123346, or with DMSO as control, and grown for the indicated periods of time in DMEM-F12 medium supplemented with 5% FBS. Then, 24 h before harvesting cells for lipidomics analysis, 10^5^ cells were seeded in 6-well plates. The day of the experiment, cells were cooled for 5 min on ice, washed twice with 2 mL of ice-cold 150 mM NH_4_CO_3_, and finally scraped in two steps in a total volume of 1.5 mL of the same buffer. Cell lysates were collected and centrifuged at 16,000× *g* for 5 min at 4 °C. Supernatants were discarded, and pellets were resuspended in 200 µL of 150 mM NH_4_CO_3_ buffer and immediately frozen in liquid nitrogen for further storage at −80 °C until lipidomes were analyzed.

***Lipidomics:*** For lipid analysis, we employed a 2-step extraction that enables efficient complex GSL detection [83]. Briefly, 1 mL of a chloroform:methanol 10:1 (*vol*/*vol*) mixture premixed with 1.4 μL of internal standard lipid mixture was added to cell lysates and subjected for 2 h at 4 °C for lipid extraction with vigorous shaking (1000 rpm in a thermomixer). The internal standard lipid mixture contained 500 pmol of Chol-d6, 100 pmol of Chol-16:0–d7, 100 pmol of DAG 17:0–17:0, 50 pmol of TAG 17:0–17:0–17:0, 100 pmol of SM 18:1; 2–12:0, 30 pmol of Cer 18:1; 2–12:0, 30 pmol of GalCer 18:1; 2–12:0, 50 pmol of LacCer 18:1; 2–12:0, 300 pmol of PC 17:0–17:0, 50 pmol of PE 17:0–17:0, 50 pmol of PI 16:0–16:0, 50 pmol of PS 17:0–17:0, 30 pmol of PG 17:0–17:0, 30 pmol of PA 17:0–17:0, 40 pmol of Gb3 18:1; 2–16:0–d9, 25 pmol of GM3 18:1; 2–18:0–d5, 25 pmol of GM2 18:1; 2–16:0–d9, and 25 pmol of GM1 18:1; 2–18:0–d5, as described elsewhere [83]. After short centrifugation, the lower organic phase was collected, and the aqueous phase was re-extracted for 1 h with 1 mL of chloroform-methanol (2:1). The lower organic phase was collected and evaporated in a SpeedVac vacuum concentrator. Each lipid extract was dissolved in 100 μL of infusion mixture consisting of 7.5 mM ammonium acetate in propanol:chloroform:methanol (4:1:2 (*vol*/*vol*)). Samples were analyzed by direct infusion in a QExactive mass spectrometer (Thermo Fisher Scientific) equipped with a TriVersa NanoMate ion source (Advion Biosciences). Then, 5 µL of sample was infused with gas pressure and voltage set to 1.25 psi and 0.95 kV, respectively.

Gb3 and Gb4 were detected in the 2:1 extract by positive ion mode FTMS as protonated ions by scanning for 30 s *m*/*z* = 800–1600 Da, at R_m/z=200_ = 280,000 with lock mass activated at a common background (*m*/*z* = 1194.81790). Every scan was the average of two micro-scans; automatic gain control (AGC) was set to 1E6 and maximum ion injection time (IT) was set to 50 ms. GM3 was detected in the 2:1 extract by polarity switch to negative ion mode FTMS as a deprotonated ion by scanning for 30 s *m*/*z* = 1100–1650 Da, at R_m/z=200_ = 280,000 with lock mass activated at a common background (*m*/*z* = 1175.77680). Every scan was the average of two micro-scans; automatic gain control (AGC) was set to 1E6 and maximum ion injection time (IT) was set to 50 ms. All data were acquired in centroid mode.

All data were analyzed with the lipid identification software LipidXplorer (LipidXplorer-1.2.8.1) [84]. Tolerance for MS and identification was set to 2 ppm. Data post-processing and normalization to internal standards were conducted manually. For the sake of simplicity, only the pertinent data are displayed (Gb3, Gb4, GM1, and GM2) and are normalized to the total lipid identified, including all phospholipids, glycerolipids, sterols, and other sphingolipids.

***dSTORM sample preparation:*** RPE-1 cells, treated as described in the “Gal3 and antibody co-binding for dSTORM analysis” section, were washed three times in PBS; a post-fixation step was performed for 15 min using PBS with 3.6% formaldehyde. The cells were then washed three times in PBS and incubated for 10 min with 50 mM NH_4_Cl, followed by three additional washes in PBS.

***Dual color dSTORM imaging:*** Fluorophores Alexa-Fluor 647 (AF647) and CF680 photo switch under reducing and oxygen-free buffer conditions, making them suitable for dSTORM single molecule imaging, which enables the localization of the emitters with sub-diffraction localization precision. Based on their close spectral proximity, AF647 was excited and acquired simultaneously with CF680 in the same dSTORM buffer (Abbelight SMART-Kit) using a 640 nm laser (Oxxius), and their respective signals discriminated after single molecule localization using a spectral demixing strategy. To implement spectral demixing dSTORM, we used a dual-view Abbelight SAFe360 equipped with two Hamamatsu Fusion sCMOS cameras and mounted on an Olympus Ix83 inverted microscope with a 100X 1.5NA TIRF objective. The SAFe360 uses astigmatic PSF engineering to extract the axial position and achieves quasi-isotropic 3D localization precision, and a long-pass dichroic mirror was used to split fluorescence from single emitters on the two cameras. Samples were illuminated in HILO at 80% of max laser power and imaged at 50 ms exposure time for 100,000 frames. Single molecule localization, drift correction, spectral demixing, and data visualization were performed using Abbelight NEO software (Acquisition: Neo_LiveAcquisition version 2.16.1, Analysis: Neo_Analysis_v38).

***Cluster segmentation:*** The PoCA (Point Cloud Analysis) software (https://github.com/flevet/PoCA (accessed on 19 March 2024)) was used and the workflow described in [85] was followed for cluster segmentation of Gal3 localizations. Briefly, the 2D co-ordinates of Gal3 localizations were loaded into the software and the following parameters were used: cut-off distance = 40 nm; minimal number of localizations = 10. The resulting clusters were categorized based on their size (nm) and area (nm^2^). The cluster size to cluster area correspondence is as follows: <100 nm = <10 (×10^3^) nm^2^; 100–300 nm = 10 to 70 (×10^3^) nm^2^; 300–500 nm = 70 to 200 (×10^3^) nm^2^; >500 nm = >200 (×10^3^) nm^2^. For statistical purposes, the range of cluster area was used.

***Spatial pattern and interaction analysis:*** The interaction analysis (probability of proximity or colocalization) was performed by loading the 2D co-ordinates of Gal3 and β_1_ integrin for each cell. The workflow for interaction analysis described in [86] was followed. The following parameters were used for computing distance distributions: Grid spacing = 0.2 (this value was chosen by sequentially reducing the grid spacing until the q (d) does not significantly change) and Kernel wt (q) = 0.001; Kernel wt (p) was used as suggested by the software. To determine the best parametric potential for the dataset, a non-parametric potential was first used to estimate the shape of interaction. The various available parametric potentials were then tested to determine the one that best fits the estimated shape of interaction. The Linear L1 potential resulted in the best fit and hence was used for subsequent datasets. The strength of interaction was plotted for each cluster type (categorized based on their area). Value = 0, weak interaction (random distribution, no colocalization); Value = 1, strong interaction (close proximity, colocalization).

***Statistical analysis:*** Experiments have been performed in biological triplicates, and 100 cells per condition were analyzed and quantified. Both *t*-test and one-way ANOVA were used for statistical significance.

## 3. Results

### 3.1. Endocytic Effects of Acute versus Prolonged Inhibition of Gal3

Gal3 affects the cell surface dynamics of α_5_β_1_ integrin [27,87]. In a pore-suspending biomembrane model, α_5_β_1_ integrin from rat liver shows fast or slow diffusivity upon addition of, respectively, low or high concentrations of oligomerization competent Gal3 [27]. It was suggested that at low Gal3 concentrations, Gal3 oligomers are nucleated on individual α_5_β_1_ integrin heterodimers, while at high Gal3 concentrations, α_5_β_1_ integrin-Gal3 assemblies may grow laterally into lattices. Consistently, it was found that membrane capacitance decreases at high Gal3 concentrations, which was interpreted as a galectin lattice-induced thickening of the membrane [27].

To explore whether such duality could also be observed on cells, we performed Gal3 binding experiments at the surface of RPE-1 cells followed by dSTORM imaging (Figure 1B, left). A recently published workflow for cluster segmentation was used [85] to identify Gal3 membrane distribution patterns. For this, we have set up a cut-off distance between two Gal3 spots equal to 40 nm and a minimum of 10 Gal3 spots as arbitrary parameters to define cluster classes. We have thereby detected different Gal3 populations at the plasma membrane, which were classified on the basis of cluster area, which varied from <10 × 10^3^ nm^2^ to more than 200 × 10^3^ nm^2^ (Figure 1B, right). Remarkably, although less frequently found, the >200 × 10^3^ nm^2^ class contained most Gal3 molecules (Figure 1C). We also performed anti-β_1_ integrin antibody co-binding, which qualitatively revealed substantial overlap with various Gal3 cluster classes, especially at the leading edge of polarized cells (Figure 1D, insets, colored arrowheads). Proximity measurements confirmed the colocalization between β_1_ integrin and Gal3 in the different clusters, compared to non-clustered Gal3 molecules (Figure 1E).

**Figure 1 biomolecules-14-01169-f001:**
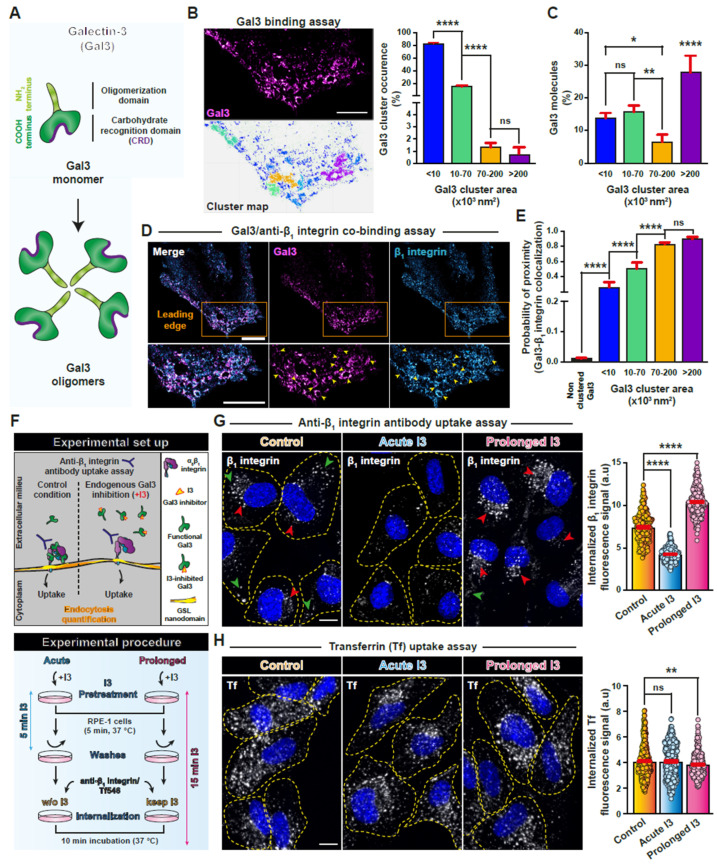
Gal3-based duality in β_1_ integrin dynamics. (**A**) Schematic representation of the molecular organization of Gal3 where the C-terminal carbohydrate recognition domain (CRD) and the N-terminal oligomerization domain are indicated. (**B**) **Left:** Representative region of interest (leading edge) of a 2D STORM image of an RPE-1 cell showing surface-bound Gal3 (**top**) and the corresponding clusters obtained after segmentation (**bottom**). **Right**: The occurrence of each type of Gal3 cluster is shown in function of their surface area (×10^3^ nm^2^). Means ± SEM, one-way ANOVA; ns = *p* > 0.05, **** *p* < 0.0001. Scale bar = 5 μm. (**C**) The percentage of Gal3 molecules is shown in function of the different Gal3 cluster populations. Means ± SEM, one-way ANOVA; ns = *p* > 0.05, * *p* < 0.05, ** *p* < 0.002, **** *p* < 0.0001. (**D**) 2D STORM image of Gal3 and anti-β_1_ integrin antibodies on RPE-1 cells (4 °C co-binding). A zoom of the leading edge is shown to illustrate the extensive level of colocalization between Gal3 and β_1_ integrin. Gal3 clusters with variable size and shape are detected (yellow arrowheads in the zoom). Scale bars = 5 μm. (E) Quantification of the probability of proximity (colocalization) between Gal3 and β_1_ integrin for each class of Gal3 clusters, compared to random non-clustered Gal3. Means ± SEM, one-way ANOVA; ns = *p* > 0.05, **** *p* < 0.0001. (**F**) Top: Anti-β_1_ integrin antibody uptake assay in RPE-1 cells. Internalized antibody is immunolabeled, imaged by confocal microscopy, and quantified. Bottom: Scheme of the protocol detailing the use of the Gal3 inhibitor I3 (10 μM) in acute versus prolonged incubation conditions. (**G**) Anti-β_1_ integrin antibody uptake assay as in (**F**). Note the shift from both peripheral (green arrowheads) and perinuclear (red arrowheads) distribution of internalized β_1_ integrin in control cells to exclusive perinuclear localization in the prolonged incubation condition. (**H**) Transferrin (Tf) internalization is very little affected by I3. In (**G**) and (**H**): Dashed lines indicate the contour of individual cells. Scale bars = 10 μm. Nuclei in blue (DAPI). Quantification of fluorescence intensities as means ± SEM, one-way ANOVA; ns = *p* > 0.05, ** *p* < 0.002, **** *p* < 0.0001.

Using an antibody uptake assay (Figure 1F, top), we assessed the role of Gal3 in the internalization of β_1_ integrin in different experimental conditions (Figure 1F, bottom). For this, we used a small molecule compound, I3 [80,81], that competitively inhibits the interaction between Gal3 and glycans. RPE-1 cells that were incubated with I3 for 5 min at 37 °C (acute protocol; see Materials and Methods for exact conditions) showed a reduction of internalized β_1_ integrin (Figure 1G, middle panel; quantification to the right). In contrast, prolonged incubation with I3 for 15 min at 37 °C resulted in an increase of β_1_ integrin internalization (Figure 1G, right panel; quantification to the right). Of note, (i) the binding efficacy of anti-β_1_ integrin antibody, i.e., the level of cell surface β_1_ integrin, remained unchanged upon I3 pre-treatment (Appendix A) and (ii), prolonged I3 incubation (15 min) decreased cellular Gal3 levels stronger than acute I3 incubation (5 min) (Appendix A).

The intracellular β_1_ integrin distribution pattern that resulted from uptake under prolonged I3 incubation conditions was different from the one observed under control conditions. In unperturbed cells, both peripheral (green arrowheads) and perinuclear (red arrowheads) accumulation patterns were observed (Figure 1G, left panel). In contrast, in cells that underwent prolonged incubation with I3, the peripheral labeling was reduced while accumulation in the perinuclear region was massively enhanced (Figure 1G, right panel).

Internalization of clathrin pathway marker transferrin (Tf) was marginally affected in these experimental conditions (Figure 1H). Our findings therefore indicate that a dual behavior in function of Gal3 activity can also be detected for endocytic β_1_ integrin uptake into cells, and that this effect is specific for this type of GL-Lect cargo protein.

### 3.2. Endocytic Effects of Acute versus Prolonged Inhibition of GSL Expression

GSLs are also part of the fabric of GL-Lect driven endocytic uptake into cells [33]. To investigate the duality aspect in this context, we have inhibited UDP-glucose ceramide glucosyltransferase (UGCG), which synthesizes the glucosylceramide on which the vast majority of GSLs are built (Figure 2A). It was previously described that GSL levels decrease proportionally to increasing times of UGCG inhibition [88]. Here, we used the UGCG inhibitor Genz-123346 to establish the inhibition kinetics on RPE-1 cells [89]. Quantification by lipidomics revealed that the cellular levels of globo-series GSLs Gb3 and Gb4 and of ganglio-series GSLs GM1 and GM2 dropped most strongly until day 3 (Figure 2B). We therefore set our experimental procedure to examine the acute impact of GSLs removal at day 3 and the prolonged effect at day 5 of Genz treatment. Endocytosis experiments were then performed at these time points (Figure 2C).

Using the anti-β_1_ integrin antibody uptake assay, we found that acute Genz treatment (3 days) induced a significant reduction of β_1_ integrin endocytosis (Figure 2D, left panel), while prolonged treatment (5 days) led to an increased uptake of the protein (Figure 2D, right panel; quantifications in histograms). Here again, the binding efficacy of anti-β_1_ integrin antibody to the cell surface was only mildly affected upon Genz treatments (Appendix A), and we documented no major alteration of β_1_ integrin cell surface levels upon GSL depletion. These results mirror those obtained with acute versus prolonged incubation of cells with I3 (Figure 1G). In further agreement with the I3 data, the peripheral (green arrowheads) and perinuclear (red arrowheads) distributions of internalized β_1_ integrin in control condition (Figure 2D, left panel) were again transformed into exclusive accumulation in the perinuclear area (red arrowheads) under prolonged GSL depletion conditions (Figure 2D, right panel).

While the clathrin pathway marker Tf was very little affected under both GSL depletion conditions (Figure 2E), Gal3 itself behaved like β_1_ integrin: acute GSL depletion induced a substantial reduction of Gal3 internalization (Figure 2F, left panel; quantification in histograms), while prolonged GSL depletion led to increased Gal3 uptake and accumulation into enlarged perinuclear endosomes (Figure 2F, right panel, red arrowheads). Similar to anti-β_1_ integrin antibody, Gal3 binding to cells was only mildly decreased upon GSL depletion (Appendix A).

The dual behavior phenotype of β_1_ integrin uptake upon I3 treatment was thereby reproduced by interfering in an acute or prolonged manner with the expression of GSLs.

### 3.3. Redistribution to EEA1-Positive Endosomes under Prolonged Interference Conditions

We then set out to further characterize the sites of intracellular cargo accumulation in the different conditions. For this, intracellular structures with internalized anti-β_1_ integrin antibodies were labeled for the early endosomal antigen 1 (EEA1). Upon acute I3 treatment, a small but significant increase of overlap was observed between internalized β_1_ integrin and EEA1 (Figure 3A, middle panel; quantification on top right). The EEA1 signal itself appeared somewhat more compacted and intense in the perinuclear region (Figure 3A, quantification on bottom right). These effects were even more pronounced upon prolonged Gal3 inhibition (Figure 3A, right panel; quantifications to the right).

The same observations for the overlap between internalized anti-β_1_ integrin antibody and EEA1 were made for the acute versus prolonged protocol in relation to GSL depletion (Figure 3B).

To further characterize the large perinuclear endosomal structures that become more pronounced under prolonged interference conditions, colocalization with EEA1 was also performed with internalized Gal3 on 5-day GSL-depleted cells (Figure 3C). The lectin strongly accumulated in the perinuclear region (as already seen in Figure 2F, right), where it increasingly colocalized with EEA1 (Figure 3C, quantification to the right).

### 3.4. Endocytic Processes under Prolonged Interference Conditions

In another series of experiments, we characterized the endocytic processes that operate under prolonged interference conditions. For this, 5-day GSL-depleted cells were incubated with the macropinocytosis marker dextran 70K (Figure 4). Macropinocytosis is a unique clathrin-independent endocytic pathway that is stimulated upon collapse of macroscale domain interactions [90]. Macropinocytosis relies on actin-driven membrane extension (ruffles) “gulping” extracellular cargoes from physiological and pathogenic sources to produce micron-size endocytic vesicles [91,92,93]. Upon prolonged GSL depletion, dextran 70K-positive macropinocytic structures appeared more abundant, bigger, and brighter when compared to the mostly discrete vesicles in control cells (Figure 4). Furthermore, the colocalization between Gal3 and dextran 70K was more pronounced in cells with prolonged GSL depletion (Figure 4, quantification to the right). These observations suggest that macropinocytic uptake is triggered upon prolonged GSL depletion.

We then analyzed whether the clathrin pathway also contributed to endocytic uptake under prolonged Gal3 or GSL inhibition conditions. For this, we used siRNAs to deplete clathrin heavy chain from RPE-1 cells, and anti-β_1_ integrin or Gal3 uptake assays were then performed in corresponding conditions. Of note, the enhanced perinuclear accumulation that correlates with prolonged inhibition (Figure 5, middle images, red and white arrowheads) was substantially decreased for all markers and all interference modalities when clathrin heavy chain was depleted (Figure 5, right images; quantifications to the right): β_1_ integrin under prolonged I3 treatment (Figure 5A), β_1_ integrin under prolonged GSL depletion (Figure 5B), and Gal3 under prolonged GSL depletion (Figure 5C).

These findings suggest that macropinocytosis and the clathrin pathway contribute to the altered endocytic uptake of β_1_ integrin that is observed upon prolonged inhibition of Gal3 or of GSL expression.

## 4. Discussion

The current study reveals that interfering with the activity of Gal3 or the expression of GSL has different outcomes in terms of endocytic uptake of β_1_ integrin into cells depending on whether the treatment with corresponding small molecule inhibitors occurs in an acute or prolonged manner. We interpret these findings in the context of the galectin lattice [23] and GL-Lect driven endocytosis models [33] (Figure 6). In the unperturbed condition, cargo glycoproteins such as β_1_ integrin can be recruited by Gal3 into galectin lattices or into tubular endocytic pits for GL-Lect driven endocytosis (Figure 6A). Upon acute interference with the dynamic Gal3 fraction at the cell surface or the acute reduction of cellular GSL levels, the GL-Lect driven de novo construction of tubular endocytic pits is readily inhibited, while preexisting lattices are more resistant, likely due to preexisting multiple bond interactions (Figure 6B). In contrast, prolonged inhibition of Gal3 or of GSL expression likely results in lattice disassembly and the release of additional cargo proteins (Figure 6C). Since GL-Lect driven endocytosis is also inhibited under these conditions, β_1_ integrin is abundantly taken up by alternative endocytic routes, i.e., the clathrin and macropinocytosis pathways.

Galectin lattices and GL-Lect driven endocytosis may thereby be viewed as intertwined processes. This interpretation is consistent with results from other recent studies. Donaldson et al. have described that for the glycosylphosphatidylinositol-anchored protein CD59, short interference with galectin activity using lactose (1h treatment) leads to an inhibition of its endocytic uptake, suggesting that this occurs via the GL-Lect mechanism [90]. In contrast, the absence of endogenous Gal3 expression using genomic depletion (Gal3-KO) resulted in increased CD59 uptake, suggesting that the protein is also associated with galectin lattices, and that upon the absence of Gal3-driven lattice assembly, it is taken up by alternative endocytic processes [90].

Dennis et al. have found that the cellular uptake of the amino acid transporter CD98/SLC3A2 is inhibited by acute incubation of cells with the Gal3 inhibitor I3, which strongly argues in favor of GL-Lect driven endocytosis of the protein [24]. In contrast, the internalization of CD98/SLC3A2 is increased upon prolonged incubation of cells with I3, while the internalized cargo is accumulated in endosomes that are patterned differently in comparison to the untreated conditions [24]. This suggests that CD98/SLC3A2 is also associated with galectin lattices and is taken up by alternative endocytic processes upon lattice disassembly.

Under prolonged interference conditions, β_1_ integrin was found to massively accumulate in the perinuclear region, where it strongly colocalized with perinuclear EEA1-positive early/sorting endosomes. This altered endocytic compartmentalization pattern, when compared to unperturbed cells, likely originates from a massive shift from GL-Lect driven endocytosis to the clathrin pathway and macropinocytosis. Accumulation in macropinosomes has previously been observed for CD59 upon prolonged lactose treatment [90], and more generally during endocytic pathway switching upon the inhibition of the clathrin machinery [94].

It seems likely that galectin lattices and GL-Lect driven endocytosis operate in an intertwined manner, the former to expand membrane residency of receptors, thereby increasing their bio-availability and functions [24,25,26,28], and the latter to dynamically localize cell surface proteins to specialized areas of the plasma membrane [95], such as the leading edge of migrating cells [96]. Lattices could also serve as ready-to-use reservoirs for membrane proteins such as integrins, which could be switched to the GL-Lect competent pool in response to physiological needs, e.g., for persistent cell migration [96]. How such switching would be operated remains to be investigated.

## Figures and Tables

**Figure 2 biomolecules-14-01169-f002:**
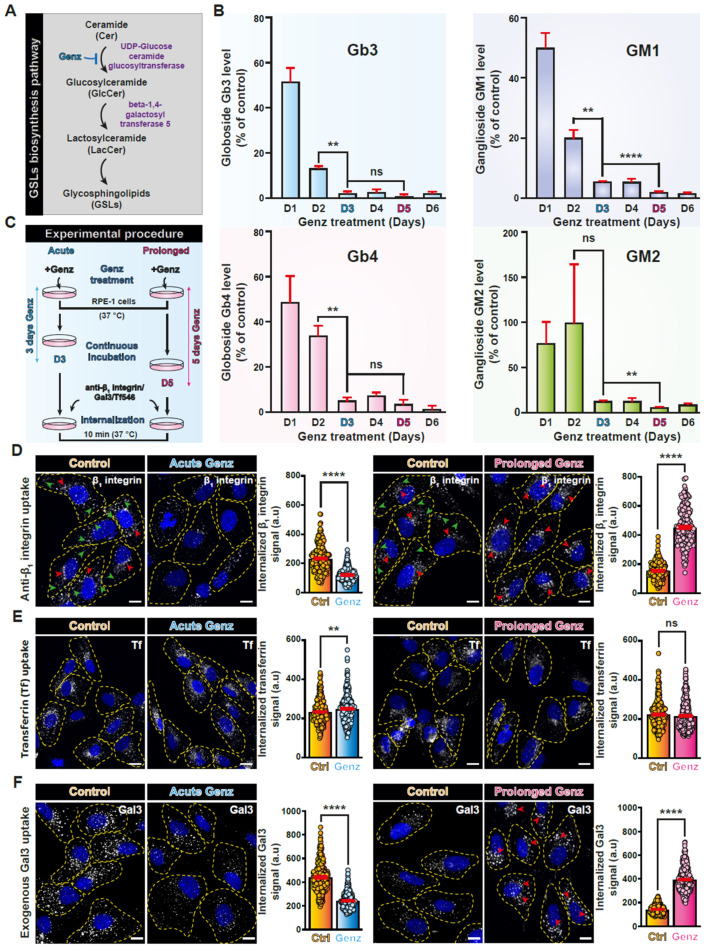
GSL-based duality in β_1_ integrin dynamics. (**A**) Simplified schematic representation of the early steps of GSL synthesis. The reaction inhibited by Genz-123346 is indicated. (**B**) Analysis of cellular levels of the indicated GSL in function of incubation time with Genz-123346. Note that the most important drop occurs up to day 3. Means ± SEM, unpaired *t*-test; ns = *p* > 0.05, ** *p* < 0.002, **** *p* < 0.0001. (**C**) Scheme of experimental procedure detailing how GSL inhibition has been set up either in acute (3 days) or prolonged (5 days) incubation conditions, prior to cargo protein internalization for 10 min. (**D**) Anti-β_1_ integrin antibody uptake assay as in (**C**). Note that β_1_ integrin uptake is inhibited upon acute Genz-123346 treatment and increased upon prolonged treatment. In the latter condition, the intracellular accumulation of β_1_ integrin is massively perinuclear (red arrowheads), compared to control cells where peripheral localizations are also observed (green arrowheads). Means ± SEM, unpaired *t*-test; **** *p* < 0.0001. (**E**) Transferrin (Tf) internalization (10 min) is only mildly affected in all conditions. Means ± SEM, unpaired *t*-test; ns = *p* > 0.05, ** *p* < 0.002. (**F**) Internalization of exogenous Gal3 (10 min). Similar to β_1_ integrin, Gal3 endocytosis is significantly inhibited upon acute Genz-123346 treatment, and increased with perinuclear accumulation upon prolonged treatment (red arrowheads). Means ± SEM, unpaired *t*-test; **** *p* < 0.0001. In (**D**–**F**): Yellow dashed lines indicate contours of cells; scale bars = 10 μm, nuclei in blue (DAPI).

**Figure 3 biomolecules-14-01169-f003:**
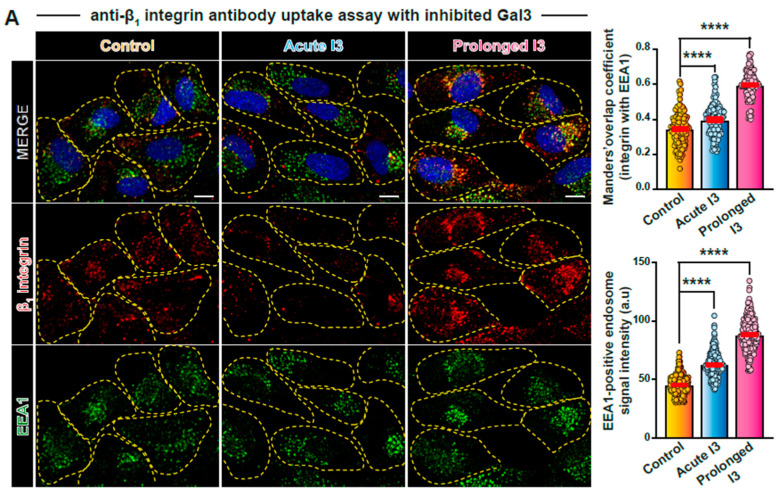
Characterization of sites of perinuclear β_1_ integrin accumulation. (**A**,**B**) Anti-β_1_ integrin or (**C**) Gal3 uptake assay (10 min) under acute or prolonged I3 (**A**) or Genz-123346 (**B**,**C**) treatment followed by immunolabeling for EEA1. The colocalization of β_1_ integrin (**A**,**B**) or Gal3 (**C**) with EEA1 as well as the fluorescent intensity of EEA1 signal were quantified (**right**). Note the increased colocalization of internalized β_1_ integrin (**A**,**B**) or Gal3 (**C**) with EEA1 and increased EEA1 signal intensity, notably in the prolonged treatment conditions. Means ± SEM, one-way ANOVA (**A**,**B**), or unpaired *t*-test (**C**); **** *p* < 0.0001. Yellow dashed lines indicate contours of cells. Scale bars = 10 μm, nuclei in blue (DAPI).

**Figure 4 biomolecules-14-01169-f004:**
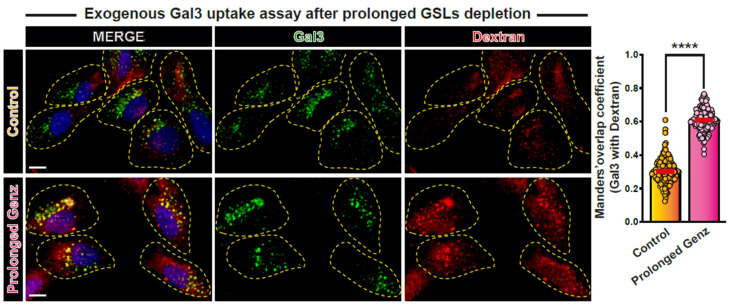
Exogenous Gal3 and dextran 70K uptake upon prolonged GSL depletion. After prolonged (5 days) treatment with Genz-123346, RPE-1 cells were continuously co-incubated (10 min) with exogenous Gal3 and dextran 70K. Note the increased perinuclear accumulation of Gal3 and its increased overlap with dextran 70K under these conditions. Means ± SEM, unpaired *t*-test; **** *p* < 0.0001. Yellow dashed lines indicate contours of cells. Scale bars = 10 μm, nuclei in blue (DAPI).

**Figure 5 biomolecules-14-01169-f005:**
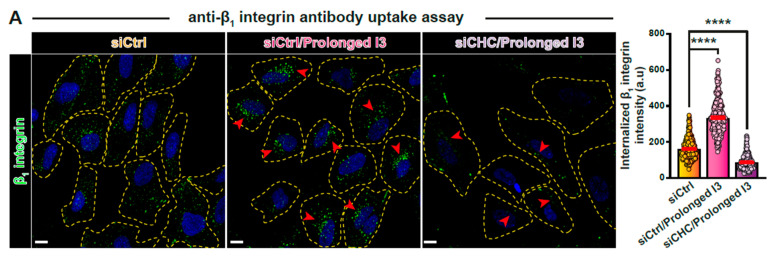
Role of clathrin in endocytic uptake under prolonged treatment conditions. (**A**–**C**) Uptake assays (10 min) of anti-β_1_ integrin antibodies (**A**,**B**) or Gal3 (**C**) upon prolonged I3 (**A**) or Genz-123346 (**B**,**C**) treatment. When indicated (siCHC), clathrin heavy chain was depleted (**right** images). The perinuclear accumulation of β_1_ integrin (**A**,**B**) or that of Gal3 (**C**) as observed in the prolonged treatment conditions (red or white arrowheads) is strongly inhibited upon clathrin depletion. Means ± SEM, one-way ANOVA; **** *p* < 0.0001. Yellow dashed lines indicate contours of cells. Scale bars = 10 μm, nuclei in blue (DAPI).

**Figure 6 biomolecules-14-01169-f006:**
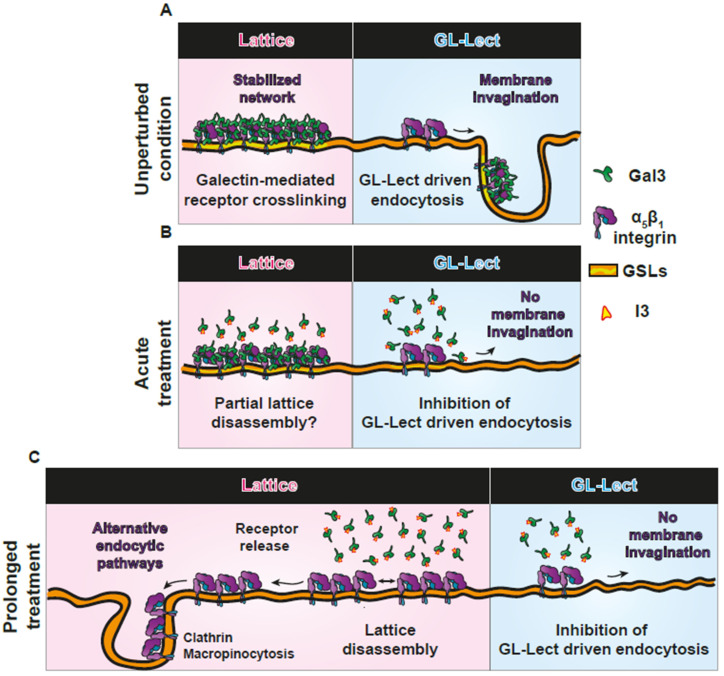
Continuum model between lattices and GL-Lect driven endocytosis. (**A**) Unperturbed condition. A glycoprotein cargo, here α_5_β_1_ integrin, is either recruited into galectin lattices (**left**, underlined in red) or internalized by GL-Lect driven endocytosis (**right**, underlined in blue). (**B**) Acute treatment conditions. Since tubular endocytic pits for GL-Lect driven endocytosis are built de novo, acute interference with Gal3 activity or GSL expression prevents their formation. In contrast, preassembled galectin lattices resist under these conditions. (**C**) Prolonged treatment conditions. Even galectin lattices are disassembled. With GL-Lect driven endocytosis being inhibited, α_5_β_1_ integrin is now internalized by alternative endocytic pathways, i.e., clathrin-mediated endocytosis and macropinocytosis.

## Data Availability

Data is contained within the article or Appendix A.

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
