# Peer review of "Exploration into Galectin-3 Driven Endocytosis and Lattices"

_biomolecules, 2024, doi:10.3390/biom14091169_

Round 1
Reviewer 1 Report
Comments and Suggestions for Authors
The work by Johannes et al. delves into galectin-mediated endocytosis processes in the context of Galectin lattices formation, in particular, the influence of the presence or not of Glycosphingolipids and of a galectin inhibitor. The results are very interesting and shed light into the so-called GlycoLipid-Lectin (GL-Lect) mechanisms, so far ill-defined.
Reviewer 2 Report
Comments and Suggestions for Authors
The primary issue raised by this reviewer regarding the manuscript by Shafaq-Zadah et al pertains to the definition of "acute" and "prolonged" inhibiting treatment. Parameters for the treatment with Genz were specified for the terms "acute" and "prolonged" as follows:
“Quantification by lipidomics revealed that the cellular levels of globo-series GSLs Gb3 and Gb4, and of ganglioseries GSLs GM1 and GM2 dropped most strongly until day 3 (Figure 2B). We therefore set our experimental procedure to examine the acute impact of GSLs removal at day 3, and the prolonged effect at day 5 of Genz treatment. Endocytosis experiments were then performed at these time points (Figure 2C).”
However, concerning the administration of I3 for extracellular galectin-3 removal, the categorization of "acute" and "prolonged" treatment seems to lack specific criteria or quantifiable measures. In addition, the authors have referenced a study by Mathew & Donaldson (reference 82) to underpin their findings:
“Line 463: Donaldson and colleagues have indeed described that for the glycosylphosphatidylinositol-anchored protein CD59, acute interference with galectin activity using lactose leads to an inhibition of its endocytic uptake, suggesting that this occurs via the GL-Lect mechanism [82].”
Nevertheless, in reference 82 (Figure 2B), the authors did not adhere to the definitions of "acute" or "prolonged" treatment. The lactose treatment (for extracellular galectin-3 removal) was administered "1 hour before beginning of the experiment and for an additional 30 minutes during the incubation with the antibody" as specified in the materials and methods section (transcribed below).
Reference 82, material and methods: For lactose treatments, a 100 mM solution of lactose (Sigma- Aldrich) in sterile complete culture medium was prepared, and the cells were incubated in the lactose solution for 1 h before the start of the experiment and during the experiment. Similarly, for sucrose treatment controls, a 100 mM solution of sucrose (Sigma-Aldrich) in sterile complete culture medium was pre- pared and used.
Given the above considerations and the significant scientific importance of the findings presented in the manuscript, it is crucial for the authors to: 1) Clearly illustrate the criteria used to determine the impact of acute and prolonged treatment. For instance, specify the extent of galectin-3 (in percentage) that was successfully eliminated from the cell surface following both "acute" and "prolonged" exposure periods; and 2) Address any inconsistencies between the current study and the references cited, particularly reference 82, to ensure coherence in the information provided.
The reviewer also noted that Figure 2F includes two controls, one for acute treatment and the other for prolonged treatment, which were not depicted in Figure 2D and E. Given that the experimental design is consistent across these figures, it was anticipated that Figures 2D and E would similarly feature controls for acute and prolonged treatment. This consistency is crucial, particularly as variations in cell responses under different treatment conditions are evident in Figure 2F.
Reviewer 3 Report
Comments and Suggestions for Authors
The manuscript: “Exploration into glycolipid-lectin driven endocytosis and galectin lattices” by Shafaq-Zadah et al., describes how competitive inhibition of Gal-3 by pharmaceutical compounds interferes with beta1 integrin antibody uptake. It is a well-written and well-illustrated manuscript with some associated pioneering scientists in the galectin-3 field. Although there was a great excitement regarding the topic and possible findings, the enthusiasm was dampened by the fact that essential controls and details are missing to derive at any irrefutable conclusions. Some of the concerns include:
- The title assumes a generalization for all Galectins, or at least multiple galectin members, yet only Galectin-3 mediated phenomena is described.
- The authors interchangeable use antibody uptake for the protein (integrin) itself.
- Although STORM is state-of-the-art, intracellular uptake is questionable in a 2D setting.
- It is unclear how specific the phenomena described is cell line dependent. Additional cell lines need to be tested.
- Galectin-3 associates itself with many cellular proteins. Is the phenomena beta 1 integrin specific or does e.g. CD44 or CD147 display similar phenotypes.
- Negative controls are missing: does omitting Galectin-3 halt Ab uptake?
- Positive controls are missing: does adding additional Galectin-3 increase Ab uptake?
- Is there a dose-response with the pharmaceutical interventions?
- The rigor and reproducibility for the experiment shown in Figure 1 A-E is not clear.
- For the experiment shown in Figure 1F-H, what would be the mechanism if Galectin-3 is washed out in both the acute (=5 mins) as well as the ‘prolonged’ (= 10 mins) exposure, yet opposite effects are seen.
- If indeed this is a time-sensitive phenomena, acute vs prolonged, multiple time points have to be tested. Where is the turning point and what would this mean in vivo?
- For Figure 1F-H is this an active process?
- How do the authors reconcile the finding in Figure1A-E regarding the importance of the leading edge, with the finding in Figure 1F-H where the leading edge does not seem as of important in that experimental set up.
- Is the leading edge phenomena associated with migrating cells, or chemotactic inducible?
- Is the leading edge phenomena a function of coating of the plate/dish/flask?
- What is the effect of I3 on Ab binding?
Round 2
Reviewer 2 Report
Comments and Suggestions for Authors
No more comments.
Reviewer 3 Report
Comments and Suggestions for Authors
The authors have addressed all my raised concerns.